# Emerging Evidences for an Implication of the Neurodegeneration-Associated Protein TAU in Cancer

**DOI:** 10.3390/brainsci10110862

**Published:** 2020-11-16

**Authors:** Stéphanie Papin, Paolo Paganetti

**Affiliations:** 1Neurodegeneration Research Group, Laboratory for Biomedical Neurosciences, Neurocenter of Southern Switzerland, Ente Ospedaliero Cantonale, Via ai Söi 24, CH-6807 Torricella-Taverne, Switzerland; stephanie.papin@eoc.ch; 2Faculty of Biomedical Neurosciences, Università della Svizzera Italiana, CH-6900 Lugano, Switzerland

**Keywords:** neurodegeneration, tauopathies, cancer, TAU protein, DNA protection

## Abstract

Neurodegenerative disorders and cancer may appear unrelated illnesses. Yet, epidemiologic studies indicate an inverse correlation between their respective incidences for specific cancers. Possibly explaining these findings, increasing evidence indicates that common molecular pathways are involved, often in opposite manner, in the pathogenesis of both disease families. Genetic mutations in the *MAPT* gene encoding for TAU protein cause an inherited form of frontotemporal dementia, a neurodegenerative disorder, but also increase the risk of developing cancer. Assigning TAU at the interface between cancer and neurodegenerative disorders, two major aging-linked disease families, offers a possible clue for the epidemiological observation inversely correlating these human illnesses. In addition, the expression level of TAU is recognized as a prognostic marker for cancer, as well as a modifier of cancer resistance to chemotherapy. Because of its microtubule-binding properties, TAU may interfere with the mechanism of action of taxanes, a class of chemotherapeutic drugs designed to stabilize the microtubule network and impair cell division. Indeed, a low TAU expression is associated to a better response to taxanes. Although TAU main binding partners are microtubules, TAU is able to relocate to subcellular sites devoid of microtubules and is also able to bind to cancer-linked proteins, suggesting a role of TAU in modulating microtubule-independent cellular pathways associated to oncogenesis. This concept is strengthened by experimental evidence linking TAU to P53 signaling, DNA stability and protection, processes that protect against cancer. This review aims at collecting literature data supporting the association between TAU and cancer. We will first summarize the evidence linking neurodegenerative disorders and cancer, then published data supporting a role of TAU as a modifier of the efficacy of chemotherapies and of the oncogenic process. We will finish by addressing from a mechanistic point of view the role of TAU in de-regulating critical cancer pathways, including the interaction of TAU with cancer-associated proteins.

## 1. Coming Together: Cancer and Neurodegenerative Disorders, Do They Share Dysregulated Pathways?

The fundamental defect resulting in cancer is an aberrant molecular machinery controlling cell division and cell death. Rather than responding appropriately to the signals that restrain cell growth, neoplastic cells divide and invade normal tissues with the potential to colonize multiple organs. In contrast, differentiated neurons display specific molecular and morphological signatures that prevent them from further cell division. However, post-mitotic neurons respond to stress conditions such as trophic factor deprivation, oxidative overload or DNA damage by up-regulating cell cycle activators, possibly causing neuronal death [1]. In fact, hallmarks of DNA replication and active cell cycle are observed in post-mitotic neurons of patients suffering of a neurodegenerative process such as in tauopathies [2,3]. This evidence conveys the postulation that neurodegeneration and cancer, despite appearing unrelated human illnesses, may both result from inappropriately regulated cellular pathways, such as cell-cycle control or cell death because of DNA damage [4,5,6].

Reinforcing this notion, an established risk factor for cancer and neurodegenerative disorders is aging—a manifestation of a time-dependent accumulation of harmful insults [7]. The two disease families share cellular and molecular hallmarks of aging [8]: genomic instability, DNA damage, epigenetic modifications, nutrient sensing abnormalities, proteostasis unbalance, mitochondrial dysfunction, telomere shortening, cellular senescence, and altered intercellular communication (Figure 1). Moreover, the aberrant regulation of common proteins and cellular pathways may occur in opposite directions. For example, whilst the regulatory mechanisms associated to the tumor suppressor P53 are frequently down-regulated in cancers [9], P53 is upregulated in concomitance to the neurodegenerative process [10,11,12]; and the reverse is true for the tumor promoting protein peptilprolyl isomerase PIN1 [13]. Both examples are discussed in more details below. These observations reinforce the concept that the occurrence of neurodegeneration and cancer may result from the deregulation of genetic factors or proteins implicated in cellular pathways common to both disease families.

Intriguing epidemiological interrelations indicate an inverse association between neurodegenerative disorders and a variety of cancer types, suggesting that a propensity for one family of diseases may decrease the risk for the other. Indeed, cancer survivors present decreased incidence for Alzheimer’s disease (AD), Parkinson’s disease (PD) and Huntington’s disease (HD), and vice versa [14,15,16,17,18,19,20,21,22]. A history of smoking related cancers has a protective impact against AD [14], whereas AD patients are less prone to develop lung cancer [23,24,25,26]. For amyotrophic lateral sclerosis (ALS) a decreased frequency of cancer is observed after disease onset [27], although a cancer diagnosis does not affect the occurrence of ALS [28,29]. In contrast, a positive correlation is observed between cancer and aging-related disorders as stroke, macular degeneration, non-neurodegenerative dementia, and osteoarthritis [15,18,19,21,30]. This is also true for the positive association of PD with melanoma and prostate cancer [30,31,32,33,34]. Cancer chemotherapies are also associated with a lower incidence of AD [35], and some of them disturb white matter structures and neuronal connectivity [36].

The interpretation of epidemiologic studies is complex and confronted with the challenge of identifying the molecular mechanisms influencing occurrence, pharmacological treatment and ultimately the survival of patients affected by one or the other of the two disorder families [37,38]. Of help is the identification of mutations in genes implicated in both disorders, such as those involved in regulation of cell cycle, DNA repair, oxidative stress, cell death and autophagy [4,39,40,41,42,43]. In this context, the protein kinase ataxia-telangiectasia mutated (ATM) and PARK2 are two examples. Germinal homozygotes mutations in ATM, a kinase tightly involved in the DNA damage response, cause ataxia-telangiectasia, a neurodegenerative disorder with a high predisposition to cancer [44]. Somatic mutations and deletions of PARK2, an E3-ubiquitin ligase involved in degradation of several target proteins including the cell cycle modulator cyclin E, have been reported in different tumor types [44], whereas germinal mutations in PARK2 are linked to PD. Large genome-wide association studies searching for co-heritability confirm shared genetic risks between AD and cancer with the largest overlap for gene sets annotated as expression regulators [45]. Interestingly, genetic components modulate the risk in the same direction and other in the opposite manner for the two disorders, but, unfortunately, it was not possible to identify in this study single nucleotide polymorphisms due to the involvement of multiple loci. Transcriptomic comparison of three cancer types and three central nervous system disorders further indicates expression deregulation in opposite directions [46]. Incidentally, genes that are strongly associated to neurodegenerative disorders, i.e., because their products are the main constituents of hallmark brain deposits and they may lead to early-onset inherited disease forms, do not exhibit typical features of oncogenes or tumor suppressors but appears to be involved in some processes associated to cancers. For example, APP promotes migration and invasion of breast cancer cells [47] and is a predictor of poor prognosis in some breast cancers [48]; whereas alpha-synuclein may be implicated in the malignant progression of meningioma [49]. A recent analysis of cancer incidence in carriers of FTDP-17 *MAPT* mutations showed increased risk of developing cancer [50]. The tumor types occurring in FTDP-17 families were variable (hematological, lung, breast, and colorectal cancers) suggesting that mutations in TAU, the protein encoded by the *MAPT* gene, may present predisposing oncogenic elements for genomic instability without tissue specificity [50]. In agreement with these data is the increased chromosomal aberration detected in lymphocytes and fibroblasts isolated from carriers of FTDP-17 *MAPT* mutations [51]. Overall, it appears that *MAPT* mutations are driving factors for neurodegenerative disorders as well as some cancer forms.

## 2. The TAU Protein

TAU is generally described as a protein highly expressed in the central nervous system. The human brain expresses at least six TAU isoforms with molecular weights ranging from 45 to 65 kDa [52,53] generated by alternative splicing of exons 2, 3 and 10 out of the 16 exons composing the *MAPT* gene [52]. The number and relative amount of the TAU splice variants vary in a cell type specific manner, during development and depending on the clinical features of neurodegenerative disorders [52,54,55]. Adding complexity, TAU proteins are modified by a considerable number and variety of posttranslational modifications; which become markedly increased in disease, as e.g., for the hyper-phosphorylated forms characterizing tauopathies [52,54,56,57]. Yet, TAU is also present in skeletal muscle, breast, kidney, prostate and in cultured fibroblasts [58,59,60,61,62,63,64,65], and at a lower level in the intestine, skin, liver, and submandibular gland [66]. A detailed analysis of the TAU species present in peripheral tissues was initially performed in rodents [67]. This led to the identification of an additional TAU isoform with a molecular weight >100 kDa (“big TAU”), generated by an unspliced 4a exon, present in rat peripheral tissues [53] and in nearly all central neurons projecting to the periphery [68]. Similar findings were reported in humans [66,69]. A detailed analysis of TAU expression at the level of mRNA, protein and post-translational modifications is crucial to better demonstrate and understand the role played by TAU in neoplastic disorders.

TAU binds to microtubules and regulates their dynamics, e.g., for the structural organization of axons and the exchange of proteins and cellular organelles between cell soma and the synapse, or for influencing the mitotic spindle. These functions are possible because TAU is a scaffold protein linking a variety of molecular partners under the control of a complex pattern of post-translational protein modifications. A simplistic concept for the role of TAU in neurodegenerative diseases is that its aberrant translational and post-translational modifications cause microtubule dissociation, followed by an increase in the soluble pool driving a toxic gain-of-function characterized by the acquisition of pathogenic conformations, self-assembly, fibril formation, and NFT deposition. This cascade of events is associated with synaptic loss, neuronal dysfunction and cell death. However, soluble TAU may relocate to other subcellular sites. In the neuronal dendrites, TAU has been shown to regulate synaptic plasticity by binding to the proto-oncogene tyrosine-protein kinase FYN, a protein involved in oncogenesis [70]. TAU is also located in the cell nucleus [71,72] and can bind DNA, acquiring DNA protecting properties [73,74] and contributing to regulate chromatin compaction [75]. Additional involvements of nuclear TAU in RNA transcription, retrotransposon mobility, and structural organization of the nucleolus and the nuclear membrane are reported [76,77,78,79]. We recently reported a modulatory effect of TAU on the tumor suppressor P53 and down-stream function such as apoptosis and senescence [80]. All these data support a role of TAU that may be independent to its binding to microtubules and may contribute to cancer. However, TAU is found predominantly bound to microtubules, and also in this function the likely contribution to cancer are well documented.

## 3. TAU and Microtubule-Targeting Chemotherapy

The mitotic spindle is the critical structure organizing the microtubule scaffold enabling chromosomal segregation and cell division. So, targeting microtubules represent a successful mode of action for cancer chemotherapy. A classic example of this class of drugs are taxanes, which bind beta-tubulin at the microtubule inner surface and inhibit microtubule depolymerization. Through the alteration of the dynamic assembly and disassembly of microtubules, taxanes restrict spindle activity and impair the cell cycle in the G1/G2 phase of mitosis. The cytostatic effect of taxanes results in the subsequent induction of apoptosis, which is partly regulated by the tumor suppressor P53 [81]. The taxane Paclitaxel present in the bark of the Pacific yew tree, is produced in a semisynthetic way from *Taxus baccata*, and is used in clinical oncology since almost three decades [82]. The resistance to taxanes observed in certain cancer types frequently limits the therapeutic efficacy. Possible causes include the action of xenobiotic efflux pumps, alterations in apoptotic and signal transduction pathways, and abnormalities in target engagement modulated by microtubule interacting proteins [83]. The microtubule-binding protein TAU may interfere with the binding of taxanes to tubulin [84]. Consequently, increased cellular concentration of TAU or its affinity to microtubules are considered factors protecting microtubules against taxane therapy [85,86,87], and are thus assessed as predictors of therapeutic efficacy for microtubule-targeting drugs [63,84,88]. For example, *MAPT* is the most differentially expressed gene as a function of response to preoperative Paclitaxel treatment in breast cancer [63], whereby low TAU mRNA predicted complete response to taxanes, as confirmed also in additional studies [85,89]. In estrogen receptor (ER)-negative breast cancer, the correlation between low TAU expression and ER status may explain the higher sensitivity to Paclitaxel [63]. Low TAU reflected by a better response to taxanes is reported also in ovarian [90,91], gastric [92], prostate [93] and non-small-cell lung cancer [94]. Notably, retinoic acid-induced TAU expression in neuroblastoma cells results in increased resistance to Paclitaxel [95], although this may be related to their differentiation state. These results feed the concept that anti-TAU drugs may be exploited as a strategy to improve the outcome of taxane-based chemotherapies. Nevertheless, some studies came to an opposite conclusion and some Paclitaxel trials did not confirm the predictive value of TAU determination [96,97,98]. The discordance between these studies may result from the choice of chemotherapy regimen, the taxane used, the cancer type, and possibly from the limitation imposed by the analysis of a single marker. Additional insights were gained by employing cellular models. Taxane-resistant prostate cells express higher level of TAU compared to parental lines, whereby TAU modulation of PI3K signaling may play a role [99]. The microRNA miR-34c-5p regulates *MAPT* gene expression in gastric cancer cell lines thereby modulating the sensitivity to Paclitaxel [100], whereas in non-small cell lung cancer cells the same effect was modulated by miR-186 [101]. The selective ER inhibitor Fulvestrant, in contrast to Tamoxifen, reduces all TAU protein isoforms and increases taxane sensitivity in ER-positive breast cancer cells [85]. It is concluded that modulation of TAU expression impacts the response to taxanes in cancer cells from diverse origins [86]. An example that qualified TAU as a potential therapeutic agent is indeed based on its microtubule-binding modulation of the mitotic spindle. The use of a tailored protein fusion between epidermal growth factor (EGF; targeting component) and TAU (effector component) resulted in a cytostatic and apoptotic response in epidermal growth factor receptor (EGFR)-positive pancreatic cancer cells [102], a finding confirmed in other models [103].

## 4. TAU as a Prognostic Marker in Cancer

The analysis of *MAPT* gene transcription and TAU protein expression in healthy and neoplastic tissues supports a role of TAU in cancer. This analytical work, in part performed in silico on available cancer databases, defines a value for TAU as a prognostic marker in various cancers (Figure 2). The following paragraphs review the outcome of these studies for distinct cancer types.

In breast cancer, higher TAU protein expression is associated to a better outcome and survival independently to the therapy [96,97,98,104,105]. However, TAU level did not correlate with tumor size or nodal status or patient age. A positive correlation between TAU expression and the receptors for estrogen and progesterone (PR) expression was confirmed in multiple studies, in particular for low grade, ER/PR-positive, and human epidermal growth factor receptor 2 (HER2)-negative cancers [96,97,98,104,105,106]. An inducible imperfect estrogen response element was identified upstream of the *MAPT* promoter [106,107,108,109,110,111,112], which is consistent with the endocrine sensitivity of TAU- and ER-positive tumors [98]. Among a panel of breast cancer cell lines with different levels of TAU mRNA and TAU isoforms, down-regulation of ER expression and the presence of ER inhibitors affected TAU expression in a cell-specific manner [85,108,113,114]. The inverse correlation TAU/HER2 is remarkable due to the proximity of the two genes in the 17q12 chromosomic region. A thorough analysis of the cancer genome atlas (TCGA) cohorts in tumors with high or low TAU expression, demonstrates a positive correlation between *MAPT* transcription and overall survival of patients with breast cancer [115]. However, a study aiming at understanding how circulating tumor cells reattach in distant tissue indicate that in metastatic breast tumor TAU is more expressed and that TAU microtubule binding is necessary and sufficient to promote tumor cell reattachment [116].

For ovarian cancer, immune histochemical analysis shows that the three-year survival was significantly higher in the TAU-negative when compared to the TAU-positive group [90]. These data suggest, in contrast to breast cancer, that high TAU expression is associated with an unfavorable prognostic. However, the results were not confirmed in the TCGA cohorts [115], which is based on gene transcript assessment rather than on protein determination. In view of the complex regulation of TAU protein homeostasis at the level of translation and post-translational modification, a careful TAU protein analysis may be more informative in this context. Notably, the endometrioid carcinoma TOV112D cells showed the highest TAU protein expression among a panel of ovarian cancer cell lines and TAU knock-down inhibited cell proliferation [91], in accordance with the favorable prognostic associated to low TAU expression [90].

An early study in prostate cancer found that TAU protein overexpression was associated with a better prognostic (lower Gleason score) in a cohort of 30 patients [117]. The use of a dephosphorylated-specific TAU antibody, demonstrated the absence of phosphorylation at the Tau-1 epitope in neoplastic prostate tissue [117]. Immune histochemical analysis on a tissue microarray containing 17,747 prostate samples showed under the selected experimental conditions detectable TAU expression in 8% of the cancer samples and no measurable TAU in the normal tissue, evidence for TAU overexpression as a moderate prognostic feature in a small prostate cancer subset [118]. TAU expression was associated with advanced tumor stage, high Gleason score, positive nodal stage, and risk for recurrence in all cancers independently of the erythroblast transformation specific-related gene (ERG) status [118]. About half of prostate cancers are due to gene fusions linking the androgen-regulated transmembrane protease TMPRSS2 with the transcription factor ERG [119,120] resulting in a massive androgen-dependent overexpression of ERG. Other somatic mutations associated to prostate cancer include *PTEN* genomic deletions, which positively associate to TAU expression with the highest *MAPT* transcription observed in ERG positive cancers. This observation is possibly linked to the suggested regulatory function in microtubule dynamics of ERG [121,122], which binds and stabilizes soluble tubulin [123]. The association between high TAU expression and poor overall survival was confirmed in an independent study [124] also describing an inverse interaction between *MAPT* and *PTEN* in prostate cancer. However, the transcriptomic-based TCGA cohorts failed to show a positive or negative association between TAU expression and survival in the prostate cancer cohort [115]. A detailed analysis of TAU in prostate cancer cell lines, revealed high expression of multiple TAU splice variants, including big TAU and a previously undescribed variant [65], in comparison to e.g., the primarily fetal TAU isoform present in human neuroblastoma SH-SY5Y cells [125,126] or the six main isoforms described in normal adult human brain [54]. Moreover, the TAU phosphorylation pattern observed in prostate cancer cells reflects what observed in tauopathies when compared to healthy adult brain with a large proportion of TAU not bound to microtubules [65]. Association of TAU to phosphoinositide 3 kinase (PI3K) suggests a microtubule-independent mechanism possibly linked to cell signaling [65,127]. Consistent with this, in docetaxel-resistant prostate cell lines [128] TAU down-regulation inhibits cell proliferation by the PI3K/mTOR signaling pathway [99].

Analysis of the bottom and top 20% MAPT expressers in pediatric neuroblastoma revealed a better prognosis for the top quintile according to the *MAPT* transcript analyzed on microarray (NCBO BioPortal) [129]. The data were substantiated with a significant correlation with apoptotic-and proliferation-linked genes. In contrast, increased survival was not associated to the mRNA for alpha-synuclein, another neurodegeneration-associated protein [129].

The value of TAU as a biomarker for disease-free survival rate in glioma (TCGA data set) was shown by comparing the bottom and top 20% *MAPT* transcript expressers [130]. Moreover, the histological tumor grade was inversely correlated with TAU expression. Consistent with these data, in the TAU mRNA-top quintile group, transcriptional activity was higher for pro-apoptotic genes and lower for proliferation-associated genes. Evidence that transcription alterations for genes associated with neurodegeneration—with the exception of *MAPT*—are not common drivers of gliomas was confirmed in another study, suggesting an important role of TAU in slowing down or preventing the clinical evolution of these tumors [131]. Histochemical analysis showed that cells from low malignancy glioma display increase TAU protein expression, with the inverse observation for cells from more aggressive tumors.

In colorectal cancer, CpG island hypermethylation in *MAPT* is found in about a quarter of the samples in a cohort with hundred stage II patients, but it was absent in normal colorectal mucosa [132]. This study was inspired by the presence of methylation in the *MAPT* promoter in AD [133], PD [134] as well as prostate cancer [135]. *MAPT* hypermethylation is a marker for lower five-year survival indicating that, similarly to breast cancer, low TAU expression is linked to a worse prognostic in both cancers. However, analysis of the TCGA database did not confirm the data [115]. At the protein level, increased TAU phosphorylation at Ser199/202 is a predictor of non-metastatic colon cancer [136]. Consistent with a main hypothesis for AD, hyperphosphorylated forms of TAU with impaired microtubule binding were reported in colorectal cell lines [137].

TAU appears implicated in Bloom’s syndrome, a rare genetic disorder resulting from homozygous mutations of the *BLM* gene with a high rate of spontaneous chromosome abnormalities and predisposition to cancer [138]. Mutated *BLM* cells experience replication stress and display chromosome segregation defects, but continue to divide indicating a tolerance for DNA damage. TAU was identified in a genome-wide RNAi screen and transcriptomic analysis as a critical protein enabling this phenotype. Indeed, TAU overexpressing Bloom’s syndrome cells undergo cell death when TAU is down-regulated [138]. This is interpreted as TAU acting as a negative regulator of DNA damage-induced cell death.

A comprehensive analysis of the TCGA cohorts shows positive association between TAU mRNA expression and survival also in kidney clear cell carcinoma, lung adenocarcinoma, pheochromocytoma/paraganglioma. In contrast, a negative association is found for colon and head and neck cancers [115].

The clinical and prognostic value of TAU analyzed at the mRNA and protein level has been investigated for many tumors with results crucially dependent on the cancer type (Figure 2). Whether the correlative studies implicating TAU in cancer will eventually demonstrate an active participation of TAU in oncogenesis requires undoubtedly further experimental evidence. As of today, the mechanisms that may explain if and how TAU differentially impact tumor cell aggressiveness in different cancer types remains at large poorly understood. As commented previously, transcriptome analysis does not take into account the pathogenic effects of protein homeostasis, which in the case of TAU is complex and tightly associated to disease. As in the case of neurodegenerative tauopathies, a detailed characterization of *MAPT* transcription and translation as well as the biochemical characterization of TAU protein including its modification, cellular distribution and interacting proteins, is now necessary in the studies linking TAU to cancer.

## 5. Possible Microtubules-Associated Mechanisms Explaining the Link between TAU and Cancer

Microtubules are the backbone on the cells and their dynamic ensure several critical functions such as cellular motility, cytoplasmic transport and cell division. Many studies suggest that microtubule dynamic is altered in cancer and linked to chromosomal instability, aneuploidy and development of drug resistances [139]. As a microtubule-associated protein, TAU expression level may interfere with several processes linking tumorigenesis and microtubules dynamic. In fact, some tumor suppressors bind to and stabilize microtubules and their inactivation may contribute to tumorigenesis through microtubules destabilization [140]. TAU may impact tumorigenesis through abnormal modulation of cell cycle progression, cell mobility or organelle organization. In fact, as mentioned previously, hallmarks of DNA replication and active cell cycle are observed in post-mitotic neurons of patients suffering of a neurodegenerative process such as in tauopathies [2,3]. Evidence that TAU may affect the mitosis process was reported using a Drosophila model, in which an excess of TAU expression induces a mitotic arrest accompanied by the presence of monopolar spindles. This mitotic defect leads to aneuploidy and apoptotic cell death [141]. TAU mutations in frontotemporal dementia cause microtubule-mediated deformation of the nucleus further resulting in defective nucleocytoplasmic transport [142], an interesting aspect as abnormal nuclear architecture is a hallmark of cancer cells [143]. Recent reports describe the implication of TAU in cell migration, a major process involved in metastasis. Using TAU shRNA in glioblastoma cell lines, the mobility of cells is strongly reduced through the Rho-associated protein kinase (ROCK) signaling pathway [144]. TAU phosphorylation status can also modulate the migration of neural stem cells [145].

## 6. Possible Microtubules-Independent Pathways Explaining the Link between TAU and Cancer

Although TAU is found almost exclusively bound to microtubules, under particular conditions TAU is also located to subcellular sites normally lacking microtubules such as the somato-dendritic compartment of differentiated neurons and the nucleus. More importantly, a presence of TAU in these peculiar subcellular sites has been associated to a role of TAU in mechanisms that are likely to be independent to its binding to microtubules. In particular, several groups have observed that TAU may participate in modifying genomic stability [51], DNA protection [73,74], and heterochromatin state [75], key processes deregulated in cancer. In this context, we address the evidence for a modulatory role of TAU in molecular pathways regulated by P53 and *BRCA1*. This chapter also review the data associating TAU to the IDH and EGFR pathways linked to glioma.

TAU may also be involved in cancer through modulation of P53. The tumor suppressor activity of the “guardian of the genome” P53 is misreguled in most cancers and may play a major role in neurodegenerative disease. Notably, whilst P53 loss-of-function is a major contributor in cancer [9], P53 expression is upregulated in AD, PD and HD [10,11,12,146]. Unusual P53 species are potential biomarkers of AD [147,148,149], the most common tauopathy with a high incidence of P53 mutations [150] and P53 deregulation [12]. Genetic alteration of P53 variants affects aging, cognitive decline, and TAU phosphorylation in mice [151,152]. Recently it has been found that P53 is part of a complex containing nuclear TAU, PIN1 and the polyA-specific ribonuclease PARN in the colon cancer cell line HCT116 [153], which are also rich in hyperphosphorylated TAU forms [154]. PARN-mediated nuclear deadenylation is activated by TAU, further potentiated by P53 and reduced by TAU phosphorylation. In this complex PARN activity targets expression of genes linked to cancer and/or AD, further supporting the functionally productive interaction of these factors in mRNA 3′-end processing in the nucleus under the modulation of TAU phosphorylation. More recently, our laboratory showed that downregulation of TAU expression impacts P53 stability in neuroblastoma cells, whereby P53 protein stabilization upon DNA damage was reduced in TAU-deficient cells. As a consequence, TAU protein depletion modifies cell fate, with decreased apoptosis counteracted by increased cellular senescence [80]. Although this role of TAU appears independent to a direct interaction with P53, it suggests that the positive association between TAU expression and cancer survival is possibly mediated by a TAU-dependent modulation of wild-type P53 stability and function. Notably a link between TAU and P53 may exist also in the context of neurodegeneration, with P53 displaying a propensity to form oligomers and fibrils upon TAU seed treatment in primary neurons, and to bind TAU oligomers in AD brain and transgenic mouse models [155]. In the same context, markers of P53-mediated response to DNA damage are reduced in AD brain. So, the current evidence indicates that TAU-deficiency as well as TAU deposition in oligomers and fibrils may contribute to an impairment of P53-mediated DNA damage response in neurodegenerative disorders and cancer.

Another possible link between TAU and cancer may appear through the BReast CAncer *BRCA1* and *BRCA2* proteins, tumor suppressors whose function is to control the integrity of the genome by promoting efficient and precise repair of double-strand DNA breaks, and mutations in these genes cause familial forms of breast, ovarian and more rarely other cancers [156,157]. A methylome profiling of AD brain, identified hypomethylation of the *BRCA1* locus, increased *BRCA1* expression and the presence of *BRCA1* in neurofibrillary tangles [158]. *BRCA1* association to fibrillary lesions is also observed in other tauopathies, namely Pick’s disease and progressive supranuclear palsy [159]. Notably, this effect is reproduced in the presence of the Abeta amyloid peptide, which causes *BRCA1* relocation to the cytoplasm and its aggregation in a TAU-dependent manner. *BRCA1* dysfunction correlates with Abeta burden and deterioration of genomic integrity and of synaptic plasticity, suggesting a disease-promoting interaction between TAU and BRCA [158]. Of possible relevance in this context, is that the DNA damage-activated checkpoint kinases Chk1 and Chk2 are able to phosphorylate TAU [160].

The mechanisms involved in TAU-associated improved survival in glioma was investigated more into details. Gliomas with isocitrate dehydrogenase (IDH1/2) mutations have a much better prognosis and response to therapy [161,162]. Notably, TAU expression is induced by mutant IDH so that TAU protein is increased in IDH1 mutated gliomas and is detected in the majority of tumor cells expressing the most common R132H IDH1 mutation. More importantly, mutant IDH enzymes favor a TAU-dependent normalization of the vasculature impairing tumor progression [131]. TAU-knockdown also slow-down migration in glioblastoma cell lines by a process that depends on the dynamics of microtubules and actin networks [144]. EGFR variants are frequently found in glioblastoma (GBM). The most common alterations are gene amplifications and rearrangements, missense mutations, and altered splicing events, which together are observed in 57% of GBMs [163]. Circumstantial evidence of a possible role of TAU in the EGFR pathway is that the activation by phosphorylation of EGFR is inversely correlated with TAU protein levels [131]. More importantly, TAU expression positively correlated with overall survival in the group of amplified wild-type EGFR GBMs, but lacked clinical relevance when combined with other EGFR variants. Mechanistically, this may be explained with the role of TAU in microtubule stabilization, whereby the presence of TAU may inhibit histone deacetylase 6 (HDAC6)-mediated acetylation of microtubule [164] and the subsequent microtubule-dependent internalization and degradation of EGFR [165]. Consistent with this, TAU overexpression in cells cause a downregulation of EGFR protein, an effect reverted in the presence of protein degradation inhibitors directed to the proteasome or lysosomal hydrolases [131].

## 7. Protein-Protein Interactions Linking TAU to Cancer

In order to unravel the role of TAU in cancer, the interaction between TAU and cancer-associated proteins will be analyzed herein. Physiological TAU is a naturally unfolded, scaffold protein, with functional domains intercalated by disordered linker sequences, similarly to other neurodegeneration-associated proteins. Beside the well-established interaction with members of the tubulin family mediated by the microtubule binding domain, TAU binds to a broad pattern of partners, including other cytoskeletal components participating to the regulation of organelle and protein transport (Figure 2) [166,167]. The function of TAU in RNA/DNA integrity (cross-reference to Colnaghi et al., same special issue) is likely to require the direct collaboration with kinases, phosphatases, chaperones and membrane proteins [55], protein families with documented tights to cancer development or suppression. The biological general repository for interaction datasets (BioGRID) interaction database reports over two hundreds TAU interactors [168]. Most relevant are considered those interactions that are confirmed by independent studies and experimental approaches, with the top five represented by glycogen synthase kinase-3beta (GSK-3beta), E3 ubiquitin-protein ligase CHIP, FYN, cyclin-dependent kinase 5 (CDK5), and the adapter protein 14-3-3zeta. In the following paragraphs we will briefly discuss the evidence linking these gene products to cancer, extending the discussion to PIN1.

The serine/threonine kinase GSK-3 was initially identified as a regulator of glycogen synthesis with follow-up evidence for participation to a wide range of cellular processes as highlighted by the identification of about hundred substrates. Aberrant GSK-3 activity is implicated in multiple pathologies including: cancer, bipolar depression, tauopathies and other neurodegenerative diseases, non-insulin-dependent diabetes mellitus and others, and is thus defined as a multitasking kinase [169]. In the context of cancer, GSK-3 functions as a tumor suppressor, e.g., when inactivated by Akt phosphorylation, or displays oncogenic properties, e.g., when stabilizing the beta-catenin complex. Consistent with this, the use of GSK-3 inhibitors remains controversial because of the ambiguous role of GSK-3 in human pathologies [170]. A complex containing TAU, cyclin-dependent kinase 5 (CDK5) and GSK-3beta is present in the brain, with CDK5 phosphorylation of TAU at Ser-235 priming further phosphorylation by GSK-3beta at Thr-231. Alternatively, CDK5-mediated phosphorylation at Ser-404 favors sequential GSK-3beta phosphorylation at Ser-400 and Ser-396 [171,172,173]. The likely contribution of this complex in TAU hyperphosphorylation implicated in neurodegenerative tauopathies suggest that a similar mechanism of protein modification may be implicated in clinically distinct disorders. In fact hyperphosphorylated forms of TAU are detected e.g., in colon cancer HCT116 cells [154] and in prostate cancer cells [65].

The serine/threonine kinase CDK5, is unique among the CDK family members in that it displays no cell cycle or mitotic function since for CDK5 no classical mediators of cell-cycle transition are known [174]. Its importance in cancer development and progression [175] is suggested by the positive correlation between high CDK5 expression and poor prognosis in pancreatic [176], lung [177], and thyroid cancer [178]. In liver carcinoma cells high CDK5 expression favors angiogenesis though hypoxia-inducible factor 1 alpha (HIF-1alpha) stabilization [179,180], and facilitating prostate cancer cell migration [181].

FYN is a non-receptor tyrosine kinase that belongs to the SRC family of non-receptor protein kinases which under normal physiological conditions is involved in signal transduction pathways in the nervous system, as well as the development and activation of T lymphocytes. The interaction between FYN and TAU is known for two decades, demonstrated by co-immune precipitation in human neuroblastoma cells and ectopic co-localization of TAU in NIH3T3 cells [182]. Whilst this interaction is expected to result in FYN-dependent tyrosine phosphorylation of TAU, the same is also important for targeting FYN to the somato-dendritic compartment where it modifies the activity of post-synaptic N-methyl-D-aspartate (NMDA) receptors and induces excitotoxicity [70,183]. In cancer, FYN contributes to the development and progression of several cancer types through the control of cell growth, death, and motility. Enhanced expression and/or activation of FYN is found in cancers of the prostate and breast, in melanoma and glioblastoma [184]. Recent studies have demonstrated the importance of FYN in the resistance or susceptibility of cancer cells to pharmacological intervention [184].

The *STUB1* encoded E3 ubiquitin ligase CHIP operates as co-chaperone in the folding, transport and degradation of proteins [185]. Taking into account the driving role of protein misfolding in many pathogenic processes including progressive neurodegenerative diseases, cancer, and a large number of rare complaints, the involvement of CHIP-mediated ubiquitination and degradation in disease is not surprising [186,187]. By assisting protein folding as a co-chaperone, CHIP is counted as a tumor suppressor [188]. Its overexpression impairs ovarian carcinoma progression [189], the growth of leukemia cells [190] and the migration and invasion of gastric cancer cells [191]. However, evidence exist of an opposite effect, where oncogenic properties are ascribed to CHIP: improved viability and accelerated tumor growth of thyroid cancer cells [192], or B-type hepatitis virus-associated carcinoma [193] are linked to CHIP overexpression. TAU is a substrate of the heat shock protein 70 (HSP70)/CHIP chaperone system, which displays homeostatic functions and the selective elimination of aberrant TAU species. Notably, CHIP presents high affinity for truncated Asp-421 TAU generated by caspase cleavage, with preferential poly-ubiquitination of this potentially pathogenic form when compared to full-length TAU. This latter demonstrated by decreased CHIP levels and increased Asp-421 TAU during AD progression [194]. TAU lesions in postmortem tissue are immune positive for CHIP, but CHIP may also accelerate TAU multimerization [195].

14-3-3zeta (also named YWHAZ) is a central hub protein for many signal transduction pathways [196]. Accumulating evidence demonstrates that it acts as an oncogene by targeting downstream protein kinases, apoptosis-associated proteins, and metastasis-related proteins in a wide range of cell activities including cell growth, cell cycle, apoptosis, migration, and invasion. It is frequently up-regulated in cancer cells possibly requiring regulation by microRNAs or long non-coding RNAs [196]. Additionally, 14-3-3zeta has shown value as a biomarker for cancer diagnosis, prognosis and chemoresistance [196]. TAU and 14-3-3zeta form a macromolecular complex [197,198,199] with GSK-3beta [200]. Moreover, 14-3-3zeta may assist the structural stability of specific TAU domains, the subcellular distribution of TAU [201], the aggregation of TAU [202,203] and ends up associated with hyper-phosphorylated TAU fibrils isolated from brains of patient with AD [204,205] or Pick’s disease [206]. Independent studies highlight high 14-3-3zeta expression in AD and Down’s syndrome brain [207] and cerebrospinal fluid [208,209].

PIN1 is the only known peptidyl-prolyl cis–trans isomerase active on the phosphorylated Ser/Thr-Pro motif. The PIN1-mediated structural conformational switch regulates at the post-translational level the function of a variety of proteins. PIN1 is therefore regulating also cellular pathways that, when dysfunctional, may lead to degenerative and neoplastic disorders. The majority of cancers present PIN1 overexpression and its down-regulation impairs disease progression, evidence for an oncogenic activity on cancer-driving pathways [210]. An opposite property appears involved in AD [211,212]. PIN1 directly binds phophoThr-231 of TAU and may act to restore its biological function on microtubules by promoting its cis/trans isomerization, its dephosphorylation and targeting to the proteasome [213,214,215,216,217,218], although the modulatory role of PIN1 on the activity TAU on microtubules was refuted in a more recent study [219]. PIN1 binding to paired-helical TAU filaments results in the depletion of soluble PIN1 that is trapped to AD neurofibrillary tangles [214]. A recent study shows that loss-of-function somatic mutations in the PIN1 gene are linked to increased TAU phosphorylation and deposition [220]. However, other studies showed that the phosphoThr231-Pro232 bond is not the preferred substrate on TAU for PIN1 [221,222] and the Ser/Thr-Pro residues appears to maintain a trans conformation when TAU is phosphorylated [223] or deposited in fibrillar structures [224].

Other TAU interacting proteins with strong relevance for cancer are the carboxyl-terminal PDZ ligand of neuronal nitric oxide synthase protein CAPON [225], the probable ATP-dependent DEAD-Box RNA helicase DDX6 [226], the proto-oncogene tyrosine-protein kinase SRC [227], the tyrosine-protein kinase ABL1 [228], the dual specificity tyrosine-phosphorylation-regulated kinase 1A DYRK1A [229], the EWS RNA-binding protein 1 (EWSR1) [230] and the sirtuin family [231].

## 8. Conclusions

It is without doubt that the main binding partners of TAU are tubulin family members. Under physiological conditions, this results with up to 90% of TAU bound to microtubules and thus not available for other interactions [232]. Accordingly, there is a consensus that TAU plays a role in modifying microtubule-targeting chemotherapeutics and, possibly, also by directly modulating microtubules and their participation to the neoplastic process. However, the binding of TAU to microtubules is highly dynamic, so that TAU is also detected in subcellular sites normally devoid of microtubules such as the nucleus or the somato-dendritic compartment of neurons. At these sites, TAU has the ability to co-localize with, and bind to, non-cytoskeletal proteins, many of which linked to cancer. These additional functions of TAU are likely to develop into relevant roles in physiological and pathological processes.

This review is an effort to compile the data supporting a role of TAU in cancer. Circumstantial evidence correlates the cellular amount of TAU protein with clinical outcomes, including survival from cancer. A better understanding of the active role of TAU in cancer will require elucidating the molecular mechanisms controlling its expression and/or the function in tumor cells or in their microenvironment. In particular, a more thorough investigation of the expression, posttranslational modification and interactions of TAU in tumorigenic tissues and cells is needed. This will certainly allow uncovering novel aspects of TAU biology that may facilitate unravelling the etiology of cancer and its relationship to neurodegenerative disorders.

## Figures and Tables

**Figure 1 brainsci-10-00862-f001:**
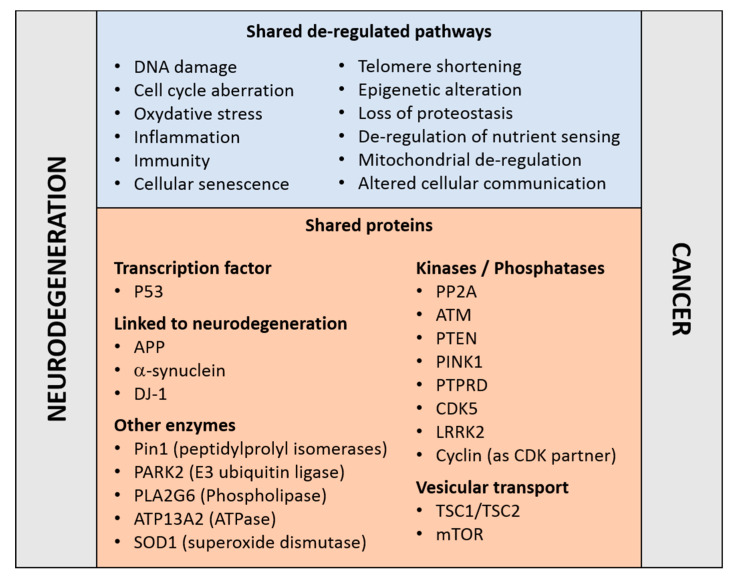
Molecular pathways (top) and proteins (bottom) that have been linked to cancer and neurodegenerative disorders, details and references are given in the main text.

**Figure 2 brainsci-10-00862-f002:**
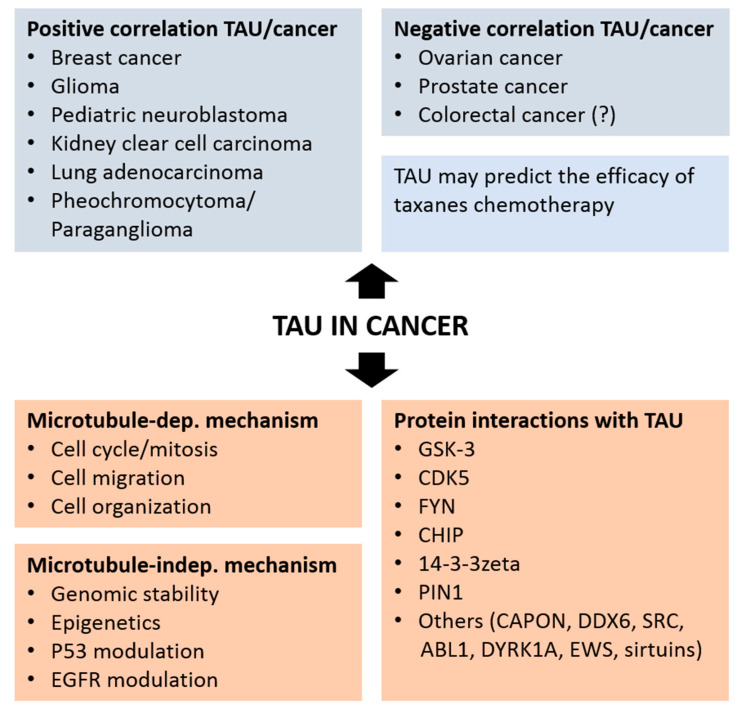
Positive and negative association of TAU expression with different type of cancers (top panels) and mechanisms and protein interactions associating TAU to cancer.

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
