# Peer review of "Emerging Evidences for an Implication of the Neurodegeneration-Associated Protein TAU in Cancer"

_brainsci, 2020, doi:10.3390/brainsci10110862_

Round 1
Reviewer 1 Report
The present review is timely as it links two important age-related diseases, cancer and Alzheimer’s disease, through the framework of the tau protein.
The first part is mostly based on clinical observations; as it is an angle I am less used to, it was refreshing to read the correlations that exist between certain cancer subtypes and the expression of tau.
As for the mechanistic details of how tau can intervene in both diseases, the review is less clear. I would for example advice the authors to cite the original references that localized tau in the nucleus (Loomis P. A., Howard T. H., Castleberry R. P., Binder L. I.(1990) Proc. Natl. Acad. Sci. 87, 8422–8426 ; Wang Y., Loomis P. A., Zinkowski R. P., Binder L. I. (1993) J. Cell Biol. 121, 257–267) as well as the work of MC Galas and L Buée who rediscovered it ten years ago. Indeed, although mcrotubule stabilization is and remains the best understood function of tau, relevant moreover for both diseases, other (in)direct functions might also come from different pools of tau.
As a second example where the review is somewhat weaker, there is the Pin1 story. After the flurry of high profile papers linking the phosphorylation-dependent prolyl cis/trans isomerase Pin1 to Alzheimer’s disease and cancer, many if not all coming from the laboratory of Prof KP Lu during the first decennium of this century, I was hoping to discover a more balanced view within the present review. According to the « cis-tauosis » hypothesis of neurodegenerative disease, whereby the cis conformation of the prolyl bond connecting pThr231-Pro232 of Tau would be a driver of disease. There is ample contradicting evidence in the literature (see below). Although this is not a review on tau and Pin1, it still might be helpful to warn the reader that the mechanistic details for most aspects remain hazy at best. This would help to research community to carefully evaluate the evidence for the « cis-tau » hypothesis and the role of the tau/Pin1 axis in AD and cancer.
In conclusion, I read with pleasure this review, and would advice the authors to deepen somewhat more he second mechanistic part .
« Cis-tauosis » contradictory evidence
- Smet & Lippens demonstrated the absence of specificity – a dipeptide was actually an excellent substrate of the enzymatic activity or binder to the WW domain. In agreement with this, Eichner & Kern showed that in full-length phosphorylated tau, the pThr231-Preo232 bond is not the preferred substrate (JMB 2016).
- The only antibody structure with a pThr231-Pro peptide of tau (Shih et al., JBC 2012) shows the bond in trans. And this antibody prefectly stains the NFTs !
- The cis-specific antibody presented by Prof KP Lu was raised against a peptide containing a chemically modified proline to induce the cis-conformer, and not against a native cis-proline.
- A careful mapping of the conformation of all prolines in tau before or after phosphorylation by CDK2 (that efficiently phosphorylates Thr231) showed that all prolines are majorly in the trans conformation (Ahuja et al . JMB 2015).
- Tau is a natively unfolded protein, so the claim that the mutation at position P301L changes the conformation of the pThr231-Pro232 bond form cis to trans needs at least some explanation.
- The initial claim that Pin1 promotes phosphorylated tau-induced MT formation in vitro has been refuted by the laboratory of Prof Kern (Kutter et al., JMB 2016).
Reviewer 2 Report
In their manuscript the authors set out to review an interesting topic of the role of Tau in neurodegeneration and cancer. Unfortunately, they fall short of this goal since the review is poorly structured and not really focused on the topic mentioned in the title, despite a vast amount of literature referenced. The discrepancy between the title and the conclusion of the article is especially confusing, since it states : “This review is an effort to compile the data supporting a role of TAU in cancer, which goes beyond microtubule binding and taxane sensitivity”. The authors should be more clear what their aim is and then make sure that this goal is met in the manuscript.
In addition, I have major concerns about this manuscript, some of which are about the form and some about the content of specific chapters.
About the form :
The purpose of a review is usually either to gather and organize information or data scattered in the published literature on a specific topic in a comprehensive and easy-to-search manner or to connect apparently unrelated publications to one another so that a new concept emerges. To fall in the first category, as the intention of the authors seem to have been, this review would need to organize better the data it is referring to. With more than 200 references, this manuscript refers to an impressive number of publications. Yet it has no picture nor table.
This manuscript is lacking an introduction clearly justifying the outlines chosen by the authors to reach the stated goal : “Elucidating how Tau is mechanistically implicated in cellular pathways common to these devastating illnesses may extend the Tau-targeting therapeutic opportunities to cancer.”
About the content :
- abstract
I feel that that the sentence “Indeed, Tau binds microtubules at the same site as taxanes” is overstated. Indeed, to that day the binding mode of tau is far from being understood, as there is no consensus on binding site for tau.
The sentence “independent studies report the association between low Tau expression and superior taxane response, the data were refuted by a meta-analysis, suggesting interference of other parameters.” Is very strong and the reader wants to know more about it; however, but this idea is not developed in the manuscript.
- Part 1 : Coming together: neurodegenerative diseases and cancer, do they share dysregulated pathways?
Authors provide a strong and interesting statement: “a rapidly expanding body of literature describes the dysregulation, often in opposite directions, of the same proteins or cellular pathways in both disorders.” Unfortunately, no details or references to this “body of literature” are given to support this statement. If it is just an introductory sentence to what will be developed later on, it should be made clear.
“A prominent example is the sharing of altered cellular functions in response to genotoxic stress, with mutations in genes involved in regulation of cell cycle, DNA repair, oxidative stress, cell death and autophagy implicated in both disorders[1-6].” This concept should be expanded with more details
The manuscript would be much more clear if all the pathways were listed in a table or better summarized in a picture showing which pathways leading to cancer are common with the ones leading to neurodegenenative disease.
Many sentences just make the reader want to know more … but there is no more : “Whilst some variants modulate in the same direction the risk for both diseases, other variants act in opposite manner.” Which ones in the same direction ? which variants act in the opposite manner ?!?
- Part 2 : Epidemiological studies associate neurodegeneration and cancer
This part addresses a very important and interesting topic, yet starting from line 85 it switches to the description of tau family and subcellular distribution of tau proteins (although interesting, this does not match the title / has nothing to do with epidemiology)
- Part 3 : TAU and microtubule-targeting chemotherapy…
“In the presence of taxanes, microtubules are frozen in stable structures due to the inhibition of the dynamic assembly and disassembly.” This is a little too simplistic. Even though high concentration of taxane in vitro do indeed freeze microtubules, taxane concentration does not need to reach that level to have an anticancer activity. Taxanes already have an effect just by perturbing the microtubule dynamicity. This should be mentioned.
Starting for line 121, there is a list of different cancers in which tau is increased or decreased? However, several major cancers are missing from this list, such as GBM (which is latter mentioned).
A table summarizing the cancers with tau alteration would significantly strengthen the manuscript.
- Part 4 : Use of TAU as a cytostatic drug
It is not clear what the goal of this very small chapter where the discussion is very short.
- Part 5 : TAU and cancer
The title of this part is not very informative since the other parts are already about tau and cancer. It starts with tau localization in cell and then switches to a cancer catalogue in no explicit order. Here, again, a table might be useful.
“As of today, the mechanisms that may explain if and how TAU differentially impact tumor cell aggressiveness in different cancer types remains at large poorly understood.”
- Part 6 : The interaction of TAU with proteins linked to cancer
It the authors want to insist upon the importance of the interplay between tau and P53 (cf abstract) they might want to make a whole section about it and a cartoon rather than hiding it in the middle of other ones.
- 7. Mechanistic evidence for a role of TAU in cancer
In this part, the authors immediately jump to EGFR and GBM without explanations. An introduction why this cancer and this pathways needs to be added. Moreover, there are many more publications discussing possible mechanistic evidence of tau in cancer
For example a recent one : Tau regulates the microtubule-dependent migration of glioblastoma cells via the Rho-ROCK signaling pathway Breuzard et al. 2019
- Conclusions
Conclusions should also be better organized and strengthened. It should only contain elements that have been discussed before, which is not the case in the current manuscript.
For example, the authors write: “However, the binding of TAU to microtubules is highly dynamic, so that TAU is also detected in subcellular sites normally devoid of microtubules such as the nucleus or the dendritic branch of neurons.” This concept is also mentioned in the introduction, even though this has not been addressed in the core of the review.
Round 2
Reviewer 2 Report
The authors have addressed most of my original concerns and added two figures that make it easier to follow.
Yet I still have an issue with the global organisation / structure of the manuscript. With the new title that focuses on cancer, the idea of interplay between pathways of cancer and neurodegenerative disorders disappears. Yet the whole first section is still about this (interesting) idea.
The more I think about it, the more I'm wandering if this section wouldn't fit better at the end of the manuscript, as a conclusion / discussion.
The structure would be the following : after having reminded the reader about tau "The TAU protein" and listed all the reasons why/how tau is related to cancer "TAU and microtubule-targeting chemotherapy", "TAU as a prognostic marker in cancer", "Mechanisms explaining the link between Tau and cancer", "Possible microtubules-associated mechanisms explaining the link between Tau and cancer", "Possible microtubules-independent pathways explaining the link between Tau and cancer", "Protein-protein interactions linking TAU to cancer" the manuscript reaches the point where the reader should be convinced of the link between tau and cancer.
Then I would come back to the fact that tau is well known for his role in ND and start discussing the possible links/interplay between the two diseases "Coming together: cancer and neurodegenerative disorders, do they share dysregulated pathways?"
I feel that it's the only way to have these 2 not so related topics to one another(yet both interesting and connected by tau) in the same review.
I would also reorganized the abstract accordingly because right now it goes back and forth between the two topics which is confusing :
While the first part line 14-19 "Neurodegenerative disorders and cancer ... risk of developing cancer." deals with the links between neurodegenerative pathways and cancer pathways (which with the new title comes as a suprise), the following line 19-28 "In addition, the expression level of TAU... processes that protect against cancer." deal with tau implications in cancer (as expected from the title) ... and then again the idea of ND vs. Cancer comes back line 28-30 : "Assigning of TAU at the interface between ... these human illnesses." before the authors switch back again to the cancer idea line 31-35: "This review aims at collecting literature data supporting the association between TAU and cancer. We will ... of TAU with cancer-associated proteins."
Author Response
Many thanks for returning in such a fast manner the manuscript after a second revision by Reviewer #2. We certainly wish to thanks warmly the Reviewer for his constructive comments, in particular during the first round of revisions that brought a strong improvement of our manuscript. Really appreciated. However, we feel that we should not follow fully the current advice from the Reviewer.
It appears that the advice is striving to have the manuscript fitting better to the title. If this is so, we would favor changing the title and revise abstract and conclusions, rather than re-organize once again the entire manuscript.
The title of the original submission was “TAU: at the interface between neurodegeneration and cancer”, and the Reviewer felt it was inaccurate because our focus was not on neurodegeneration. We agreed that this needed to be improved. In the revised version we proposed “TAU: emerging evidence for an implication in cancer”, and the Reviewer proposes now to move the first chapter describing the connections between neurodegeneration and cancer to the end of the manuscript. We partly disagree with the Reviewer because in our opinion, the chapter describing the link between cancer and neurodegeneration is the optimal introduction to the review, which has the objective to describe the evidences that TAU is involved in cancer, in addition to its crucial role in neurodegeneration (not detailed in this review). The objective is not to use Tau as an introduction to highlight the communalities between the two disease families, as the Reviewer is suggesting doing.
Nevertheless, the comments made by the Reviewer prompted us to suggest few modifications
- Title: “Emerging evidence for an implication of the neurodegeneration-associated protein TAU in cancer”. The previous title was “TAU: Emerging Evidences for an Implication in Cancer” In this manner both disease families are mentioned and this allows us to start describing the link between the two disorders.
- Abstract, we moved the sentence “Assigning TAU at the interface between cancer and neurodegenerative disorders, two major aging-linked disease families, offers a possible clue for the epidemiological observation inversely correlating these human illnesses” from the end of the paragraph to lines 20-22. We revised the last sentence, lines 32-36, which is now reading “This review aims at collecting literature data supporting the association between TAU and cancer. We will first summarize the evidence linking neurodegenerative disorders and cancer, then published data supporting a role of for TAU as a modifier of the efficacy of chemotherapies and of the oncogenic process. We will finish by addressing Then, we will address from a mechanistic point of view the role of TAU in de-regulating critical cancer pathways, including the interaction of TAU with cancer-associated proteins.”
- Conclusions, we modified the sentences at lines 495-502 to “This review is an effort to compile the data supporting a role of TAU in cancer, which goes beyond microtubule binding and taxane sensitivity. Circumstantial evidence correlates the cellular amount of TAU protein with clinical outcomes, including survival from cancer. As an example, high TAU expression is linked to slower disease progression in gliomas and breast carcinoma, pointing to a tumor suppressor role in these types of cancers. A better understanding of the Direct evidence for an active role of TAU in cancer will require elucidating the molecular mechanisms controlling its expression and/or the function in tumor cells or in their microenvironment. In particular Despite initial evidence, a more thorough investigation of the expression, posttranslational modification and interactions of TAU in tumorigenic tissues and cells is needed.”
We hope that our revisions are acceptable and take the opportunity to communicate our best regards